# Does a National Park Enhance the Environment-Friendliness of Tourists as an Ecotourism Destination?

**DOI:** 10.3390/ijerph18168321

**Published:** 2021-08-06

**Authors:** Eunseong Jeong, Taesoo Lee, Alan Dixon Brown, Sara Choi, Minyoung Son

**Affiliations:** 1Department of Tourism Management, Honam University, Gwangju 62399, Korea; eunseong@honam.ac.kr; 2Department of Geography, Chonnam National University, Gwangju 61186, Korea; 3Department of Economics, Chonnam National University, Gwangju 61186, Korea; alan@jnu.ac.kr; 4Graduate Program of Tourism Studies, Honam University, Gwangju 62399, Korea; myvenus73@gmail.com; 5Department of Hotel and Airline Service, Cheongam University, Suncheon 57997, Korea; miin0202@nate.com

**Keywords:** ecotourism, new ecological paradigm, environmental-friendly behavior, conservation

## Abstract

Governments have designated national parks to protect the natural environment against ecosystem destruction and improve individuals’ emotional and recreational life. National parks enhance environment-friendly awareness by conducting ecotourism activities and individuals with environment-friendly awareness are inclined to continue to visit national parks as ecotourism destinations. The New Environmental Paradigm (NEP) is a widely used measure of environmental concern, suitable for measuring the environment-friendly attitude and revisit intention of visitors of national parks. Therefore, the study carried out structural equation modeling (SEM) to investigate the relationship between the NEP, national park conservation consciousness and environment-friendly behavioral intention. Based on the results, an implication is presented to induce national parks to cultivate individual environment-friendly awareness and for visitors to pursue sustainable, environment-friendly tourism behavior. The findings indicate that national parks are to expand educational programs and facilities for eco-tourists visiting national parks to maintain a balanced relationship between themselves and nature and have a strong environmental awareness to preserve the natural environment.

## 1. Introduction

Over the past several decades, the tourism industry has generated positive ripple effects, such as job creation, regional economic revitalization and expansion of social infrastructure in tourist destinations [1]. It has also resulted in negative effects, such as conflict between tourists and local residents, crime and gentrification [2]. Ecosystem destruction caused by development results in citizen antipathy. Consequently, policies and programs have been proposed to minimize the destruction of the natural environment by tourism development. Ecotourism has emerged and is frequently mentioned by tourism studies as a vehicle to minimize the damage to the natural environment caused by tourism development. However, instances have occurred of ecotourism deviating from its original purpose of responsible tourism, whereby tourists actively participate to strengthen environmental awareness and protect the global environment and endangered flora and fauna, and degenerating into mass tourism [3,4].

Ecotourism has often been criticized as only satisfying the requirement of tourists and the need to promote strong policies and programs to preserve the natural environment has been emphasized [5,6,7]. From this point of view, even before the advent of ecotourism, many countries designated national parks to protect natural or historical resources. Yellowstone National Park was designated in the United States as the world’s first national park in 1872 in order to protect its natural resources and allow individuals to enjoy them. Since the creation of Yellowstone National Park, most countries have designated national parks to protect natural resources and to improve individuals’ recreation and emotional life [8]. Korea designated Mt. Jiri as its first national park in 1967, with 22 national parks covering 6726 km^2^ designated as of 2021 [9].

The Korea National Park Service, which is tasked with managing national parks in Korea, operates programs to promote the use of national parks as ecotourism destinations by the public and facilities are provided to raise the environmental awareness of visitors to national parks. Previous studies have identifed that national park visitors have a higher awareness of environment-friendly awareness compared to visitors of other tourist destinations [10,11]. Studies demonstrated that national parks are able to enhance environment-friendly awareness by conducting ecotourism activities in national parks and individuals with environment-friendly awareness are inclined to continue to visit national parks as ecotourism destinations.

Based on studies demonstrating that national park visitors have a higher environment-friendly awareness than visitors of other tourist destinations, the New Ecological Paradigm (NEP) is a scale for measuring the environment-friendly attitude and revisit intention of visitors of national parks. The NEP was first introduced by Dunlap and Van Liere as a set of beliefs about an individual’s environment expressed through an ecological worldview outside the dominant social paradigm [12]. The NEP scale has been recognized as a reliable quantitative multi-item scale for measuring an individual’s beliefs about the natural environment and is used in disciplines such as politics [13], psychology [14] and sociology [15]. However, in tourism studies, few studies have applied the NEP as a measure of tourists‘ environment-friendly beliefs.

In other words, tourism studies have mainly adopted scales and models to investigate tourists’ environment-friendly awareness through simple measurement of environmental knowledge [16], planned behavior theory with environmental attitudinal factors [17] and transformative theory [18]. Whereas those studies measured tourists’ fractional attitude and knowledge level of environment friendliness, the NEP observes individuals’ holistic perspectives of environment-friendly philosophies.

The study, therefore, uses the NEP scale to establish revist intention of visitors to national parks as ecotourism destinations based on their environment-friendly awareness. The study investigates the relationship between the NEP, national park conservation consciouness and behavioral intention. Based on the results, a plan is presented to induce natural parks such as national parks to cultivate individual environment-friendly awareness and tourists to pursue sustainable eco-friendly tourism behavior.

## 2. Conceptual Background and Hypotheses Development

### 2.1. Nature-Based Tourism

Nature-based tourism (NBT) has gradually increased in popularity and is the fastest growing sector within tourism across Asia and elsewhere [19,20]. With the increased popularity and demand for NBT, tourism sites have been challenged to sustain natural resources, while also optimizing tourist activity and satisfaction [21]. Scholars have pointed out that NBT is an alternative tourism which includes adventure tourism, sustainable tourism, ecotourism and even cultural tourism [22,23]. Studies have defined NBT as tourists’ activities which mainly depend on natural resources [24,25]; moreover, it is a significant means for individuals to recuperate from mental fatigue and stress through the direct enjoyment of untouched natural environments [26,27].

Previous studies have indicated that NBT provides both environmental perspectives for tourists and environmental sustainability for the local communities [28,29,30]. In other words, NBT fosters appreciation of nature and the ecosystem of both tourists and local communities; furthermore, it is a form of sustainable development which minimizes environmental and social impacts.

### 2.2. Ecotourism and National Park

Although ecotourism has been defined in many different ways, the concept of ecotourism is suggested, by previous studies, as a means for sustainable development and a specific tourism activity in which individuals and small groups of tourists visit natural areas as an educational method [31,32,33]. Ecotourism is a tourism activity in which tourists experience the uncontaminated and pristine natural ecology without consuming it and enjoy the natural and socio-cultural characteristics of the region. Ecotourism is based on sustainable tourism that enables tourism activities to take place within the natural ecosystem by mobilizing devices that limit negative effects on the natural environment [34]. Ecotourism also contributes to the conservation and sustainable use of ecological resources, revival of the local economy, enhancement of tourists’ understanding of the environment and ecosystem conservation [35,36].

From this point of view, ecotourism plays an important role in nurturing citizens who have a pro-environmental attitude to both solve environmental problems and maintaining a healthy balance of life. Governments support ecotourism activities for individual ecological experiences, the designation of ecological parks for environmental conservation education, the establishment of facilities and the protection and preservation of ecological parks [37,38]. In this context, national parks are being developed and operated as representative ecotourism destinations [39]. National parks not only have outstanding natural environments and historical and cultural value, but are also characterized as public goods in which the state regulates resources at an appropriate level [40].

Countries operate ecotourism experience programs to diversify experiential activities beyond simple enjoyment of the natural environment in national parks, to raise awareness of the value of national parks through experiences [41,42,43]. The operation of ecotourism experience programs in national parks is shifting from a management method to a different direction, whereby changes towards environment-friendly attitudes are induced through education and experience reinforcement for national park visitors [44]. As such, changes in the management policies of national parks play a role in strengthening the environment-friendly attitude of national park visitors and allowing them to develop a new paradigm for the environment.

### 2.3. New Ecological Paradigm

The NEP scale is a frequently used measure of environmental concern. It has been shown to display reliability and validity as a measurement instrument [45,46], which displays internal consistency in both national and international studies [47]. The scale was originally known as the New Environmental Paradigm scale [12], but was revised by Dunlap and his colleagues [48] to become the New Ecological Paradigm scale. 

The NEP scale has been used in previous studies on the value of environmental goods and services in terms of willingness to pay [49,50]. These studies demonstrate that those with a strong environment-friendly attitude place increased value on environmental goods and services. Furthermore, environmental attitudes and nature-based tourism motivations have a close positive relationship [51]. The NEP scale has also been applied to different demographic groups, such as children [52], ethnic minorities [53], college students [54] and leisure activity participants [55] and across different countries [56]. The largest criticism of the NEP scale regards its dimensionality. Its original conception as a unidimensional instrument [12] has been empirically challenged by studies which show it to be multidimensional [57]. Some caution is suggested in applying the scale as a measure of a unidimensional construct [58]. Research also suggests that, while the NEP scale is a valid construct of environmental perceptions, it should not be considered as an all-encompassing measure of them [59]. 

Although previous literature raised caution regarding the NEP scale, partially modified versions of the NEP scale have been applied in many studies, which have affirmed the scale to be a concise, valid and reliable measure of an individual’s world-view or attitude toward the environment. In particular, the NEP scale rejects the idea that nature has no value except for human use and emphasizes that the modern industrial society has exceeded its ecological limits and is disrupting the ecosystem [60]. The NEP scale most strongly emphasizes a world-view explaining the relationship between humans and the environment, that of an awareness of the human environment. Instruments appropriate to each region and object have been developed for the NEP scale. This study adopts four of the five sub-factors suggested by Dunlap et al. [49], namely, the reality of limits to growth, the fragility of nature’s balance, anti-exceptionalism and the possibility of an eco-crisis.

### 2.4. Environmental Conservation and Environment-Friendly Behavioral Intention

Nature provides positive benefits to humans, such as promoting biodiversity, promoting mental stability, strengthening communities and providing positive economic benefits [61]. However, rapid industrialization and urbanization have damaged natural ecosystems, resulting in a decrease in the socioeconomic effects that nature provides to humans [62]. To solve this problem, governments, NGOs and international organizations are operating policies and programs to expand protected natural areas and promote environment-friendly thinking of citizens [63]. Nature conservation includes elements of sustainable production and multi-purpose use and not only it helps to create economic benefits for the community, but has also positive effects, such as protecting natural resources and endangered animals [62].

As interest in nature conservation increases, ecotourism allows visitors to experience natural ecology with relatively minimal damage to the natural environment caused by tourism development [64]. Environment-friendly behavior is a consumption behavior for environmental conservation, not for personal gain, but for society as a whole and it is a consciousness for the conservation of natural environment, knowledge and natural resources [16,65]. However, in the real world, it is difficult to expect environment-friendly behavior from only voluntary individual actions due to the social dilemma that arises between individual convenience and welfare and the interests of society as a whole [66]. 

Unlike other behaviors, environment-friendly behavior contributes to the welfare of society as a whole, rather than only the consumer’s own benefit [67]. Moreover, environment-friendly behavior is characterized by human consumption of goods and services. This behavior can be defined as consumer behavior that considers both the satisfaction of personal needs and the survival of all mankind with a conscious and consistent interest, so that individual actions in the process of selection, use and disposal positively impact the environment [68,69]. From this perspective, the study defines environment-friendly behavior as individual and collective actions and sequences of actions taken by individuals to solve environmental problems or issues in their daily lives and tourism activities as the NEP proliferates in society.

## 3. Methods

### 3.1. Research Model and Hypotheses

The study designed a research model by reviewing theories and studies on the relationship between the NEP scale, environment conservation consciousness and environment-friendly behavior perceived by Korean national park visitors (See Figure 1). Whereas previous studies on the NEP were oriented to individuals’ holistic environment-friendly perspectives and behaviors according to the NEP, we focused on their direct consequences, specifically, environment-friendly attitudes and behavioral intentions. Studies identified that the NEP had a significant effect on environment conservation consciousness [70,71]; moreover, the NEP also had a significant effect on environment-friendly behavioral intention [72,73,74]. Research demonstrated the relationship between environment conservation and environment-friendly behavioral intention, which indicated individuals with higher scores on environment conservation expected to have higher scores of environment-friendly behavioral intention [75]. Based on previous research, the present study proposes the following hypotheses and research model (See Figure 1).

**Hypothesis** **1 (H1).**
*Reality to limits of growth of the NEP will have a positive and significant effect on environmental conservation.*


**Hypothesis** **2 (H2).**
*Fragility of nature’s balance of the NEP will have a positive and significant effect on environmental conservation.*


**Hypothesis** **3 (H3).**
*Anti-exceptionalism of the NEP will have a positive and significant effect on environmental conservation.*


**Hypothesis** **4 (H4).**
*Possibility of an eco-crisis of the NEP will have a positive and significant effect on environmental conservation.*


**Hypothesis** **5 (H5).**
*Reality to limits of growth of the NEP will have a positive and significant effect on environment-friendly behavioral intention.*


**Hypothesis** **6 (H6).**
*Fragility of nature’s balance of the NEP will have a positive and significant effect on environment-friendly behavioral intention.*


**Hypothesis** **7 (H7).**
*Anti-exceptionalism of the NEP will have a positive and significant effect on environment-friendly behavioral intention.*


**Hypothesis** **8 (H8).**
*Possibility of an eco-crisis of the NEP will have a positive and significant effect on environment-friendly behavioral intention.*


**Hypothesis** **9 (H9).**
*Environmental conservation will have a positive and significant effect on environment-friendly behavioral intention.*


### 3.2. Study Setting

The study was undertaken among Korean visitors who visited three national parks in southwestern Korea. The three national parks, Naejangsan, Mudeungsan and Wolchulsan National Parks, are mountainous parks with very little commercial development and have a reputation as ecotourism destinations. The Korea National Park Service provides environmental education programs and facilities to visitors in order to conserve ecosystems and strengthen visitors’ environment-friendly perception and behavior. Figure 2 provides the research area of the study.

### 3.3. The Sample and Data Collection

The study focused on Korean visitors who had visited Naejangsan, Mudeungsan, or Wolchulsan National Parks. The three national parks were selected as they actively provided environmental education programs to visitors and local communities in order to strengthen their environment-friendly perspectives and behavior by allowing them to enjoy a variety of leisure activities in a national park [76]. This study conducted a field survey of a random sample of individuals who visited the national parks with the primary purpose of enjoying ecotourism activities. All survey participants were informed that the survey data would be confidential and destroyed after conducting the data analyses of the study. Due to the ongoing COVID-19 pandemic, activities in national parks were restricted. The restrictions excluded any on-site questionnaires being administered; therefore, the study conducted an online self-administrated questionnaire which could advantageously replace an on-site questionnaire by simultaneously increasing response rate and reducing branching error rates [77,78]. Consequently, self-administrated structured questionnaires were completed online by the subjects to obtain data on the parks from 16 April to 15 October 2020. A total of 1025 questionnaires were returned and, after a preliminary check, 952 were deemed usable for statistical analysis.

### 3.4. Measures and Data Analysis

The questionnaire asked the survey participants about their perception of the NEP scale, environmental conservation and environment-friendly behavioral intention; moreover, it contained questions to gather demographic data. The 12 items NEP scale, developed from Ntanos et al.’s [50], Harrison’s [79] and Luo and Deng’s [51] studies, asked about four sub-dimensions of the NEP, the reality to limits of growth, fragility of nature’s balance, anti-exceptionalism and possibility of an eco-crisis. Environment conservation was assessed using four items related to nature conservation, which were adapted from Corral-Verdugo et al.’s [71] and Baral’s [80] research, reflecting the singular aspect. Environment-friendly behavioral intention was measured using four items adapted from Gao et al.’s [16] and Brick and Lewis’s [81] studies (See Table 1). All items were measured on 5-point Likert scales ranging from (1) “strongly disagree” to (5) “strongly agree”.

The data were analyzed utilizing IBM SPSS 25.0 Windows and AMOS 25.0. All items of the NEP, environmental conservation and environment-friendly behavioral intention were first transformed into z-scores, which allowed the study to calculate the probability of a score occurring within a standard normal distribution. Then, the study conducted exploratory factor analysis (EFA) and Cronbach’s alpha test for purification of scale items. After purifying the scale items, the study conducted confirmatory factor analysis (CFA) to ensure validity and reliability; thereafter, it conducted structural equation model to test hypotheses of this research.

## 4. The Results

### 4.1. Sample Characteristics

The majority of survey participants were female (52.3%, 498 people), while 47.7% (454 people) were male. The greatest proportion of participants, 35.2% (335 people), were in their 20s; 25.4% (242 people) were in their 30s; 19.2% (183 people) were in their 40s; 14.9% (142 people) were in their 50s; and 5.2% (50 people) were 60 years or older. As for the educational background of the respondents, 45.6% (434 people) were enrolled or were going to enroll at 4-year colleges, 32.9% (313 people) were enrolled at high school, 17.0% (162 people) were enrolled or were going to enroll at 2-year colleges and 4.5% (43 people) were enrolled or were going to enroll at graduate school. In terms of visit experience in a national park, 73.8% (704 people) had visited a national park more than five times, 25.9% (247 people) had visited a national park from two to four times and 0.3% (three people) were first-time visitors to a national park.

### 4.2. Purification of Scale Items

There were two steps for scale purification of the 20 items of the dimensions in the study. The first step was to conduct the Cronbach’s alpha test, which evaluates reliability of the measured items in terms of internal consistency by testing correlations between items. The alpha value of all items ranged from 0.805 to 0.950, which satisfied the recommended standard of 0.7. The second step was to conduct EFA. EFA identifies the hypothetical factor structure of a dimension and it demonstrates the dimensionality of a modified structure of a construct when the items of latent structure are rephrased or modified, not only from one population to another, but also from one discipline to another.

EFA of the 20 items was conducted through principal component factor analysis, utilizing Varimax rotations to identify the dimensionality of the NEP, environment conservation and environment-friendly behavioral intention, with six factors explaining 81.446% of the variance. The Kaiser-Meyer-Olkin (KMO) was 0.932, which exceeded the recommendation value of 0.6, and Bartlett’s test of sphericity was statistically significant (χ^2^ (190) = 15,989.269, *p* < 0.001). All the items related well to their dimensions, with factor loadings being greater than 0.5, which was expected, as all were well-established scales.

In addition, the Cronbach’s alpha test was checked in order to analyze the reliability of the six dimensions extracted. All dimensions were judged to be reliable, with a recommened value of 0.7 or higher, including 0.950 for reliability to limits of growth, 0.935 for environment conservation, 0.932 for environment-friendly behavioral intention, 0.920 for fragility of nature’s balance, 0.870 for possibility of an eco-crisis and 0.805 for anti-exceptionalism (See Table 2).

### 4.3. Validity and Reliability of Scales through Confirmatory Factor Analysis

CFA was used to examine the psychometric properties of our scales. Table 3 shows the proposed measurement model for conducting CFA, which consisted of 6 factors and 20 observed variables. A CFA with robust maximum likelihood estimation was conducted on the variables of the RLG, FNB, AE, PEC, EC and EFBI, utilizing AMOS 25.0. In step one, CFA for the proposed measurement model with the major dimensions was tested. The proposal model produced a clearly good fit with data, the chi-square value for the overall model fit was significant, χ^2^ (132) = 341.529, *p* < 0.001, CMIN/DF = 2.587, TLI = 0.981, NFI = 0.979, CFI = 0.987, GFI = 0.967 and RMSEA = 0.041. As such, the model was acceptable for testing the hypotheses of this study.

Both validity and reliability of the measurement model were assessed. The reliability test was performed by calculating composite reliability, which tested the internal consistency of the observed variables measuring each dimension. The composite reliability test provided an estimate of the variance shared by the respected indicators, through the item loadings obtained within a nomological network [82]. Composite reliability of the scales ranged from 0.844 to 0.956, which indicated good internal consistency of the model (C.R. > 0.7). Average variance extracted (AVE) was utilized to examine discriminant validity to show that measurement items in the model not only were pure measures of discrete traits but also had method variance [83]. The AVE of the six dimensions exceeded the recommended standard of 0.50. Standardized factor loadings for items ranged from 0.668 to 0.922. The ideal factor loading is greater than 0.70 [84], but studies suggest that at least 0.50, for standardized factor loading, is acceptable [85].

The study tested discriminant validity of the proposed model by using correlation and AVE values. Discriminant validity hypothesizes that items will correlate more strongly among themselves than they will to items from other dimensions that are theoretically not supposed to correlate [86]. Table 4 presents that the coefficient of determination of all dimensions (0.131~0.650) was smaller than AVE (0.644∼0.847); thus, discriminant validity was confirmed.

### 4.4. Hypothesis Tests

Figure 3 depicts the hypotheses testing results. The hypothesized model for the study showed a good model fit, with CMIN/DF = 2.679, GFI = 0.966, NFI = 0.978, TLI = 0.980, CFI = 0.986 and RMSEA = 0.042. The proposed impact of the NEP on environment conservation was tested (H1, H2, H3 and H4). H1 tested the relationship between reality to limits of growth and environmental conservation, which was not supported (β = 0.012, *p* = 0.835). H2 examined the relationship between fragility of nature’s balance and environmental conservation, which was supported (β = 0.283, *p* = 0.000). H3 tested the relationship between anti-exceptionalism and environmental conservation, which was supported (β = 0.326, *p* = 0.000). H4 tested the relationship between possibility of an eco-crisis and environmental conservation, which was supported (β = 0.203, *p* = 0.000). Next, the proposed impact of the NEP on environment-friendly behavioral intention was examined (H5, H6, H7 and H8). H5 tested the relationship between reality to limits of growth and environment-friendly behavioral intention, which was not supported (β = 0.039, *p* = 0.518). H6 examined the relationship between fragility of nature’s balance and environment-friendly behavioral intention, which was supported (β = 0.149, *p* = 0.036). H7 tested the relationship between anti-exceptionalism and environment-friendly behavioral intention, which was not supported (β = 0.109, *p* = 0.087). H8 tested the relationship between possibility of an eco-crisis and environment-friendly behavioral intention, which was not supported (β = 0.070, *p* = 0.302). Lastly, H9 investigated the relationship between environmental conservation and environment-friendly behavioral intention, which was supported (β = 0.278, *p* = 0.000). The conceptual model of this study, therefore, was partially accepted.

## 5. Discussion and Implications

Nations have paid attention to ecotourism development in protected areas such as national parks in order to provide not only nature-based tourism activities, but also strengthen awareness of environmental conservation among individuals. The existing studies in the tourism field have largely focused on tourist activities of ecotourism; on the other hand, few studies have identified environmental consciousness through participating in ecotourism. Many disciplines have actively studied individuals’ environmental awareness as our society has faced an environmental crisis. Accordingly, scholars have pursued the development of a reliable and valid measurement to evaluate individuals’ environmental attitudes. The NEP is widely used as a dimensional measurement of environmental attitudes by measuring the relationship between humans and nature [87]. Based on the above, the primary purpose of the present study was to investigate the relationship between the NEP, environmental conservation and environment-friendly behavioral intention perceived by visitors to national parks in South Korea.

A total of four underlying dimensions (i.e., reality to limits of growth, fragility of nature’s balance, anti-exceptionalism and possibility of an eco-crisis) informing national park visitors’ perceived performance of environmental perspectives were derived based on a review of existing studies. The analysis results demonstrated that three dimensions of the NEP, fragility of nature’s balance, anti-exceptionalism and possibility of an eco-crisis were statistically significant for environmental conservation of the visitors, while only one dimension, reality to limits of growth, was not. Furthermore, the analysis results identified that only one dimension of the NEP, fragility of nature’s balance, was statistically significant for environment-friendly behavioral intention of national park visitors, while three dimensions of the NEP were not. Lastly, environmental conservation of the visitors was statistically significant for environment-friendly behavioral intention. Accordingly, the present study has the following theoretical and practical implications for policy-makers.

First, previous ecotourism studies have conducted research on ecotourism motivations, development strategies, plans, visitor management and values. This study applied the NEP scale to investigate the relationship between eco-tourists’ environment-friendly awareness and behavior by examining the relationship between the awareness of national park visitors of environmental conservation and their environment-friendly behavior in national parks. It differs from previous studies, in that the NEP scale was applied to ecotourism research and presented its reliability and validity. In particular, although the management and operation of national parks is relatively active in Korea, compared to other Asian countries, research on national parks has been mainly limited to the collection of basic data on the status of the ecological environment, consequently, research centered on environmental conservation. To simultaneously raise visitors’ ecotourism activities and environmental conservation awareness, this study can provide theoretical implications for future ecotourism-related research by examining the relationship between environmental awareness and environment-friendly behavioral intentions in national parks.

Second, the study analysis of the relationship between the NEP scale and environmental conservation showed that visitors who were highly aware of the fragility of nature’s balance, anti-exceptionalism and the possibility of an eco-crisis had a higher awareness of environmental conservation in national parks. There is, therefore, a need to expand educational programs and facilities to raise awareness of an environmental crisis. Moreover, based on the result that the reality of limits of growth did not have a significant effect on environmental conservation, it is necessary to develop educational programs that awaken visitors to national parks to the finite nature of the natural environment.

Third, among the dimensions of the NEP scale perceived by visitors to national parks, only fragility of nature’s balance had a significant effect on environment-friendly behavior intention. This indicates that visitors are aware that the destruction of the natural environment can occur even in national parks, if they are not managed in an environmentally friendly way. Accordingly, recognizing that conservation of the natural environment is an important purpose of national park designation, national park management agencies should establish management and planning to carry out ecotourism activities in national parks only within the scope in which environmental conservation can be achieved.

Fourth, the study confirmed that the visitor awareness of the environmental conservation of national parks has a significant effect on environment-friendly behavioral intention. The results suggest that the environment-friendly behavior of visitors can be elicited by expanding programs and facilities, which increase awareness of environmental conservation. Therefore, the national park management agency should be responsible for protecting the ecological environment of the national park through publicity and facility expansion, thereby influencing visitors to voluntarily follow its environmental conservation policy, allowing the park to play a role as an eco-tourism site.

## 6. Conclusions

The pursuit of convenience by exploiting the natural environment has resulted in its damage and some flora and fauna facing extinction. As individual tourism activities increase, the destruction of the natural ecosystem by tourism development has led to the emergence of an alternative form of tourism, ecotourism, although it does not perfectly preserve the natural environment. Accordingly, the government is establishing policies and operating facilities to simultaneously pursue tourism activities for tourists and preserve of the natural environment through the designation and management of national parks. Previous studies suggested that tourists who enjoy eco-tourism have a higher awareness of environmental conservation, compared to tourists who pursue mass tourism. Based on this, the study measured the environmental conservation consciousness of visitors to national parks. The study empirically analyzed the relationship between the NEP scale, environmental conservation and environment-friendly behavior intention. The results suggested the necessity of national parks to expand educational programs and facilities for eco-tourists visiting national parks, maintain a balanced relationship between themselves and nature and have a strong environmental awareness to preserve the natural environment.

Although this study presents theoretical and practical implications, it has the following limitations. First, due to the COVID-19 pandemic, activities in national parks were restricted. Consequently, data for the empirical analysis were collected through an online survey and the survey participants had a limited understanding of the study. Furthermore, there were many outliers in the data and more hypotheses proposed in the conceptual model of this study were rejected compared to the results of previous studies. Therefore, data for future research are to be collected through field survey rather than online survey. Second, many national parks implement educational programs to improve environment-friendly awareness and a large number of visitors participate in these programs. However, this study used random sampling to extract samples from national park visitors. Accordingly, future research should limit the sample to visitors who participate in a national park’s environment-friendly education program in order to investigate the relationship between the environment-friendly awareness of national park visitors and their participation in eco-tourism.

## Figures and Tables

**Figure 1 ijerph-18-08321-f001:**
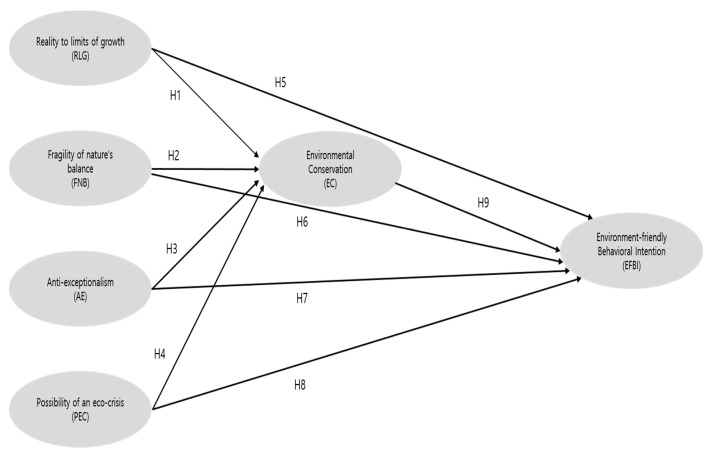
Conceptual model for hypotheses of the study. Note: H = hypothesis.

**Figure 2 ijerph-18-08321-f002:**
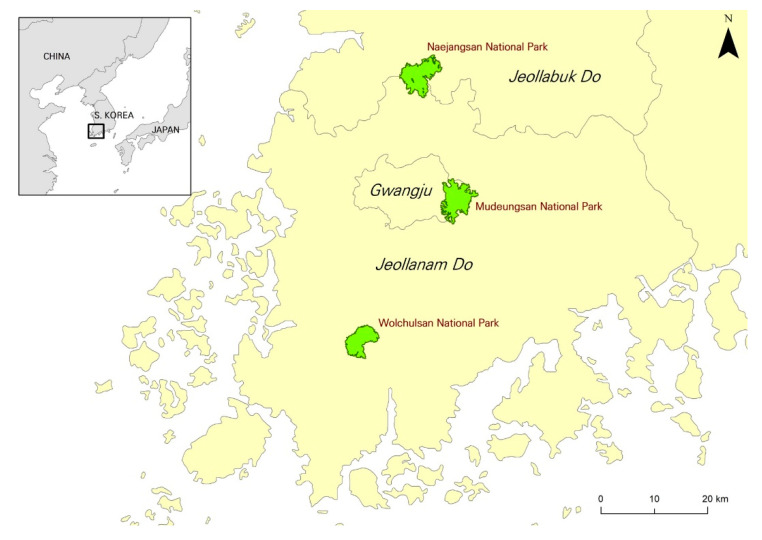
The research site: Three national parks in southwestern Korea.

**Figure 3 ijerph-18-08321-f003:**
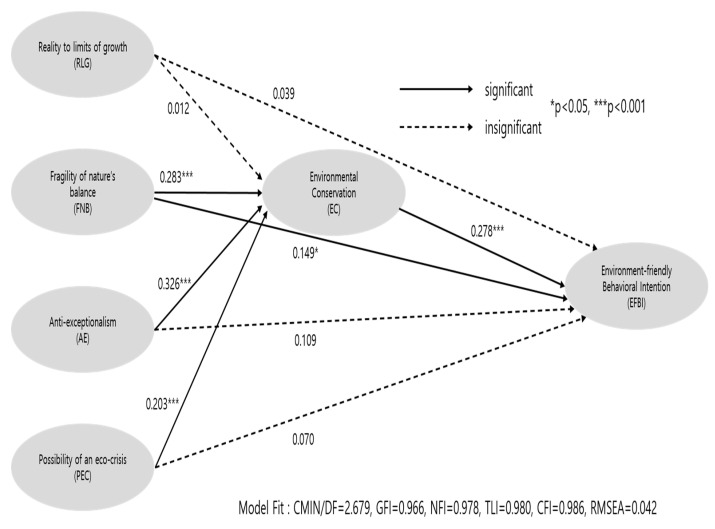
Structural equation model results.

**Table 1 ijerph-18-08321-t001:** Measures of the NEP, environmental conservation and environment-friendly behavior intention.

Factor		Item
NEP	Reality to limits of growth (RLG)	Humans are approaching the limits of the capacity of people the Earth can support (RLG1). The Earth provides plenty of natural resources for humans if we learn how to develop and protect them (RLG2). The Earth is like a ship with very limited natural resources (RLG3).
	Fragility of nature’s balance (FNB)	When humans abuse nature, it often produces disastrous consequence (FNB1).
		The balance of nature is strong enough to cope with the impacts of powerful nations (FNB2). The balance of nature is delicate and easily destroyed (FNB3).
	Anti-exceptionalism (AE)	The present human development of natural resources will ensure that we do not make the Earth unlivable (AE1). Despite humans’ special abilities, we are still subject to the laws of nature (AE2). Humans will ultimately be sufficiently knowledgeable about how nature works to be able to control it (AE3).
	Possibility of an eco-crisis (PEC)	Humans are heavily abusing the environment (PEC1). The ecological crisis facing humankind has been greatly progressed (PEC2).If humans continue with their present environmental abuse, we will experience a major ecological catastrophe (PEC3).
Environmental Conservation (EC)		I am willing to accommodate closing trails in the national park (EC1). I am able to help maintain the quality of the national park (EC2). I am willing to donate to protecting the national park (EC3). I can inform other of the significance of national park as an environment-friendly tourism destination (EC4).
Environment-friendly Behavioral Intention (EFBI)		I am willing to revisit this national park as an ecotourism destination (EFBI1). I am willing to recommend this national park as an ecotourism destination to others (EFBI2). I am willing to introduce this national park as an ecotourism destination to others (EFBI3). I am willing to visit other national park (EFBI4).

**Table 2 ijerph-18-08321-t002:** Exploratory factor analysis and reliability of analysis results.

Dimension	Items	Factor Loading	Eigenvalue	Variance (%)	Alpha
RLG	RLG2	0.823	3.499	17.493	0.950
	RLG1	0.796			
	RLG3	0.654			
FNB	FNB1	0.753	3.429	17.145	0.920
	FNB2	0.730			
	FNB3	0.692			
AE	AE1	0.812	2.560	12.799	0.805
	AE2	0.773			
	AE3	0.583			
PEC	PEC1	0.792	2.414	12.070	0.870
	PEC2	0.757			
	PEC3	0.692			
EC	EC1	0.829	2.227	11.135	0.935
	EC2	0.820			
	EC3	0.811			
	EC4	0.804			
EFBI	EFBI1	0.903	2.161	10.804	0.932
	EFBI2	0.882			
	EFBI3	0.858			
	EFBI4	0.855			
Kaiser-Meyer-Olkin measure of sampling adequacy = 0.932, Bartlett’s test of sphericity: χ2 = 15,989.269, df(p) = 190(0.000).Total variance explained by 6 factors: 81.446%.					

**Table 3 ijerph-18-08321-t003:** Results of confirmatory factor analysis.

Dimension	Items	Standardization Coefficient	Variance of the Error	AVE	C.R.
RLG	RLG1	0.891	0.145	0.761	0.904
	RLG2	0.879	0.158		
	RLG3	0.668	0.330		
FNB	FNB1	0.844	0.218	0.847	0.943
	FNB2	0.922	0.102		
	FNB3	0.911	0.112		
AE	AE1	0.752	0.353	0.644	0.844
	AE2	0.716	0.370		
	AE3	0.806	0.233		
PEC	PEC1	0.817	0.229	0.769	0.908
	PEC2	0.892	0.124		
	PEC3	0.768	0.265		
EC	EC1	0.922	0.116	0.826	0.950
	EC2	0.919	0.119		
	EC3	0.880	0.160		
	EC4	0.812	0.262		
EFBI	EFBI1	0.876	0.156	0.845	0.956
	EFBI2	0.915	0.106		
	EFBI3	0.879	0.149		
	EFBI4	0.877	0.164		
χ^2^/df = 2.587, TLI = 0.981, NFI = 0.979, CFI = 0.987, GFI = 0.967, RMSEA = 0.041.					

**Table 4 ijerph-18-08321-t004:** Correlation and discriminant validity tests.

Factors	RLG	FNB	AE	PEC	EC	EFBI
RLG	**0.761**					
FNB	0.711 **	**0.847**				
	(0.506)					
AE	0.714 **	0.749 **	**0.644**			
	(0.509)	(0.561)				
PEC	0.746 **	0.806 **	0.686 **	**0.769**		
	(0.557)	(0.650)	(0.471)			
EC	0.586 **	0.694 **	0.613 **	0.659 **	**0.826**	
	(0.343)	(0.482)	(0.376)	(0.434)		
EFBI	0.362 **	0.456 **	0.407 **	0.423 **	0.479 **	**0.845**
	(0.131)	(0.208)	(0.166)	(0.179)	(0.229)	

** *p* < 0.01; bold represents AVE; parenthesis contain the coefficient of determination.

## Data Availability

Not applicable.

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
