# Peer review of "Does a National Park Enhance the Environment-Friendliness of Tourists as an Ecotourism Destination?"

_ijerph, 2021, doi:10.3390/ijerph18168321_

Round 1
Reviewer 1 Report
- L15-18, the paper proposed NEP as a research scale or a method at first, and then mentioned the relationship between NEP, national park conservation consciousness and environment-friendly behavioral intention. The logic of this part is not clear enough, I recommend developing a little more about this.
- L 68-73, overall research innovation, especiallythe significance of innovation compared with current studies, should be emphasized more clearly.
- For 3.1, Why these variables were selected and the improvements compared to previous literature should be clarified more clearly
- A compass should be added in Fig.2.

Author Response
We thank you for giving of your time and appreciate your expert guidance. We have read your comments and revised our text to reflect you as follows.
- 15-18, the paper proposed NEP as a research scale or a method at first, and then mentioned the relationship between NEP, national park conservation consciousness and environment-friendly behavioral intention. The logic of this part is not clear enough, I recommend developing a little more about this.
: The New Environmental Paradigm (NEP) is a widely used measure of environmental concern, suitable for measuring the environment-friendly attitude and revisit intention of visitors to national parks. Therefore, the study carried out Structural Equation Modeling (SEM) to investigate the relationship between NEP, national park conservation consciousness, and environment-friendly behavioral intention.
- 68-73, overall research innovation, especially the significance of innovation compared with current studies, should be emphasized more clearly.
: In other words, tourism studies have mainly adopted scales and models to investigate tourists’ environment-friendly awareness through simple measurement of environmental knowledge [16], planned behavior theory with environmental attitudinal factors [17], and transformative theory [18]. Whereas the studies measured tourists’ fractional attitude and knowledge level of environment friendliness, the NEP observes individuals’ holistic perspectives of environment-friendly philosophies.
- For 3.1, Why these variables were selected and the improvements compared to previous literature should be clarified more clearly
: Whereas previous studies on the NEP were oriented to individuals’ holistic environment-friendly perspectives and behaviors according to the NEP, we specifically focused on their direct consequences, specifically environment-friendly attitudes and behavioral intentions.
- A compass should be added in Fig.2.
Figure is in the file.
Reviewer 2 Report
This is a very interesting and well-developed paper. However, there are some issues the authors should revise:
- the introduction should expand the statement ‘in tourism studies, few studies have applied the NEP as a measure of tourists‘ environment-friendly beliefs’ and present previous research to justify the research gap.
- in the theoretical background, the authors should add a sub section about nature-based tourism.
- the method should provide references to explain why these three national parks are the most suitable for the study. The authors should also develop the rationale of using self-administrated structured questionnaires.
Author Response
We thank you for giving of your time and appreciate your expert guidance. Authors have read your comments and revised our text to reflect you as follows.
- The introduction should expand the statement ‘in tourism studies, few studies have applied the NEP as a measure of tourists’ environment-friendly beliefs’ and present previous research to justify the research gap.
: In other words, tourism studies have mainly adopted scales and models to investigate tourists’ environment-friendly awareness through simple measurement of environmental knowledge [16], planned behavior theory with environmental attitudinal factors [17], and transformative theory [18]. Whereas the studies measured tourists’ fractional attitude and knowledge level of environment friendliness, the NEP observes individuals’ holistic perspectives of environment-friendly philosophies.
- In the theoretical background, the authors should add a sub section about nature-based tourism.
: We added a sub session as below.
2.1. Nature-based Tourism
Nature-based tourism (NBT) has gradually increased in popularity and is the fastest growing sector within tourism across Asia and elsewhere [19,20]. With the increased popularity and demand for NBT, tourism sites have been challenged to sustain natural resources while also optimizing tourist activity and satisfaction [21]. Scholars have pointed out that NBT is an alternative tourism which includes adventure tourism, sustainable tourism, ecotourism, and even cultural tourism [22,23]. Studies have defined NBT as tourists’ activities which mainly depend on natural resources [24,25]; moreover, it is a significant means for individuals to recuperate from mental fatigue and stress through the direct enjoyment of untouched natural environments [26,27].
Previous studies have indicated that NBT provides both an environmental perspectives for tourists and environmental sustainability for the local communities [28-30]. In other words, NBT fosters appreciation of nature and the ecosystem of both tourists and local communities; furthermore, it is a form of sustainable development which minimizes environmental and social impacts.
- The method should provide references to explain why these three national parks are the most suitable for the study. The authors should also develop the rationale of using self-administrated structured questionnaires.
: First, we provided a statement “The study focused on Korean visitors who had visited Naejangsan, Mudeungsan, or Wolchulsan National Parks. The three national parks were selected as they have actively provided environmental education programs to visitors and local communities in order to strengthen their environment-friendly perspectives and behavior by enjoying a variety of leisure activities in a national park.”
Second, we added the rationale of using self-administrated structured questionnaires “Due to the ongoing COVID-19 pandemic, activities in national parks were restricted. The restrictions excluded any on-site questionnaires being administered. Consequently, Self-administrated structured questionnaires were completed by the subjects to obtain data on the parks from 16 April to 15 October 2020.”

Round 2
Reviewer 2 Report
While the authors have revised the paper, they have not enough improved the method. They should also provide references to justify the selection of the national parks and the use of the self-administrated structured questionnaires.
